# Factors associated with disease severity and mortality among patients with COVID-19: A systematic review and meta-analysis

**Vignesh Chidambaram**[1], **Nyan Lynn Tun**[1], **Waqas Z. Haque**[1], **Marie Gilbert Majella**[2], **Ranjith Kumar Sivakumar**[3], **Amudha Kumar**[4], **Angela Ting-Wei Hsu**[1], **Izza A. Ishak**[1], **Aqsha A. Nur**[1], **Samuel K. Ayeh**[5], **Emmanuella L. Salia**[6], **Ahsan Zil-E-Ali**[1], **Muhammad A. Saeed**[7], **Ayu P. B. Sarena**[8], **Bhavna Seth**[9], **Muzzammil Ahmadzada**[7], **Eman F. Haque**[10], **Pranita Neupane**[5], **Kuang-Heng Wang**[1], **Tzu-Miao Pu**[1], **Syed M. H. Ali**[11], **Muhammad A. Arshad**[12], **Lin Wang**[1], **Sheriza Baksh**[1], **Petros C. Karakousis**[5], **Panagis Galiatsatos**[9]*

1 Johns Hopkins Bloomberg School of Public Health, Baltimore, Maryland, United States of America, 2 Department of Preventive and Social Medicine, Jawaharlal Institute of Postgraduate Medical Education and Research, Puducherry, India, 3 Department of Anaesthesia and Intensive Care, Prince of Wales Hospital, The Chinese University of Hong Kong, Shatin, Hong Kong, China, 4 Department of Internal Medicine, University of Arkansas for Medical Sciences, Little Rock, Arkansas, United States of America, 5 Division of Infectious Diseases, Department of Medicine, Johns Hopkins School of Medicine, Baltimore, Maryland, United States of America, 6 Department of Pediatrics, Johns Hopkins School of Medicine, Baltimore, Maryland, United States of America, 7 Johns Hopkins University, Baltimore, Maryland, United States of America, 8 Bhayangkara Setukpa Hospital, Sukabumi, Indonesia, 9 Division of Pulmonary and Critical Care Medicine, Department of Medicine, Johns Hopkins School of Medicine, Baltimore, Maryland, United States of America, 10 Southern Methodist University, Dallas, Texas, United States of America, 11 Fatima Memorial Hospital, Lahore, Pakistan, 12 Nishtar Hospital, Multan, Pakistan

* pgaliat1@jhmi.edu

**Data Availability Statement:** All relevant data are within the manuscript and its Supporting information files.

## Abstract

### Background

Understanding the factors associated with disease severity and mortality in Coronavirus disease (COVID-19) is imperative to effectively triage patients. We performed a systematic review to determine the demographic, clinical, laboratory and radiological factors associated with severity and mortality in COVID-19.

### Methods

We searched PubMed, Embase and WHO database for English language articles from inception until May 8, 2020. We included Observational studies with direct comparison of clinical characteristics between a) patients who died and those who survived or b) patients with severe disease and those without severe disease. Data extraction and quality assessment were performed by two authors independently.

### Results

Among 15680 articles from the literature search, 109 articles were included in the analysis. The risk of mortality was higher in patients with increasing age, male gender (RR 1.45, 95%

**Funding:** The authors received no specific funding for this work.

**Competing interests:** The authors have declared that no competing interests exist.

CI 1.23–1.71), dyspnea (RR 2.55, 95%CI 1.88–2.46), diabetes (RR 1.59, 95%CI 1.41–1.78), hypertension (RR 1.90, 95%CI 1.69–2.15). Congestive heart failure (OR 4.76, 95%CI 1.34–16.97), hilar lymphadenopathy (OR 8.34, 95%CI 2.57–27.08), bilateral lung involvement (OR 4.86, 95%CI 3.19–7.39) and reticular pattern (OR 5.54, 95%CI 1.24–24.67) were associated with severe disease. Clinically relevant cut-offs for leukocytosis(>10.0 x10$^9$/L), lymphopenia(< 1.1 x10$^9$/L), elevated C-reactive protein(>100mg/L), LDH(>250U/L) and D-dimer(>1mg/L) had higher odds of severe disease and greater risk of mortality.

## Conclusion

Knowledge of the factors associated of disease severity and mortality identified in our study may assist in clinical decision-making and critical-care resource allocation for patients with COVID-19.

## Introduction

Since the first documented reports of the severe acute respiratory syndrome coronavirus 2 (SARS-CoV-2) infection, the virus has had a global impact, affecting millions, which led the World Health Organization (WHO) to declare the outbreak a pandemic [1, 2]. Patients who develop Coronavirus Disease 2019 (COVID-19) may require hospitalization and intensive care unit admission [3–5].

With variable access to critical care resources across countries, recent guidelines for the COVID-19 pandemic have called for allocating life sustaining treatments based on a patient's risk of mortality [6, 7]. Health systems preparedness requires a deeper understanding of how to effectively triage patients with COVID-19, in order to maximize the benefit of scarce intensive care unit resources while minimizing the potential harm of outpatient management of ill patients. While the guidelines warrant utilizing triage scores that have been previously validated for assessing organ failure/dysfunction and survival (e.g. sequential organ failure assessment, SOFA), these are non-specific in etiology [7, 8]. Thus, having an understanding of the predisposing conditions and disease-specific clinical, laboratory and radiological parameters, may lay the groundwork for developing a COVID-19 specific composite score at a later stage, which can predict unfavorable clinical outcomes.

With the massive influx of studies on COVID-19 in the recent months and their often conflicting or unclear findings, a systematic review of the factors associated with survival or disease severity in patients with COVID-19 that takes into consideration the inherent variability in study population, will be of great utility to clinicians, researchers and policy makers. In this systematic review and meta-analysis, we sought to better understand the clinical, laboratory and radiological parameters associated with mortality and disease severity among patients with COVID-19.

## Methods

### Search strategy and study selection

We followed the PRISMA guidelines for reporting in systematic reviews and meta-analyses [9]. We searched PubMed, Embase and the WHO COVID-19 database by using the search strategy included in the supplementary document in S1 Appendix (Section I). For PubMed and Embase, an initial search on March 26, 2020 was conducted, and updated multiple times

with the final update performed on May 7, 2020. The WHO database was initially downloaded on March 27, 2020 and the final update was performed on May 8, 2020. Only articles published in the English language were included. We only included article published in peer-reviewed academic journals; we did not include articles uploaded in the preprint servers, as they are not peer reviewed and the findings may not be reliable [10].

We included observational studies that included patients with microbiologically confirmed SARS-CoV-2 infection, irrespective of the age of the participants. The study designs of the included studies were assessed and recorded independently by two authors (VC and MM) acting as arbiters. The differentiation between case series and cohort studies was made based on the criteria outlined by Dekkers et al [11]. Case reports, case series, and randomized control trials were excluded from this review. We included all studies that reported a direct comparison of clinical, laboratory or radiologic characteristics between a) patients who died and those who survived or b) patients with severe disease and those without severe disease. Only those studies which defined "severe disease" based on the American Thoracic Society guidelines for the treatment of Community-acquired Pneumonia [12] or the Chinese National Health Commission guidelines for the Treatment of Novel Coronavirus infection [13], were included in our analysis. Studies which only described the characteristics of patients who died or patients with severe disease were excluded if there was no comparison group.

We only included studies reporting primary hospital data on patients, while studies with centralized data from national health agencies and databases were excluded from our review. Efforts were made, as feasible, to minimize overlap of patients across studies by collecting information on the name of the hospital, date of hospital admission of the participants and the names of the investigators.

## Literature screening

The COVIDENCE platform was used for conducting this systematic review [14]. After de-duplication, the titles and abstracts of the articles retrieved from the search strategy were screened independently by at least two of the following authors (VC, NT, WH, RS, AK, AH, II, AN, SA, ES, and MS) and conflicts were resolved by consensus between VC and NT. The full texts of the articles included after title and abstract screening were independently screened by at least two of the following authors (VC, NT, WH, RS, AK, AH, II, AN, SA, ES, and MS) and conflicts were resolved based on consensus between VC and NT. Specific reasons for study exclusion are listed in Fig 1.

## Data extraction and quality assessment

Data extraction was performed independently by at least two of the authors (VC, NT, WH, MM, RS, AK, AH, II, AN, SA, ES, MS, AS, KW, and TP) and conflicts were resolved by a consensus between two authors (MM and VC). The data extraction form for this review was created using the Qualtrics platform [15]. The primary outcomes were a) death of the patient and b) the presence of severe disease in the patient. There were no secondary outcomes. Data on the study characteristics, source of funding, demographic characteristics, comorbidities, clinical symptoms, in-hospital complications, laboratory, and radiological features of the study participants were extracted. Data on sex of the patient, smoking status, presence of comorbidities, clinical symptoms, in-hospital complications, and radiological features were extracted as binary variables, while age was extracted as continuous data. For the laboratory parameters, data were extracted as both continuous and categorical variables. Binary data for the laboratory variables were collected using all the cut-offs described in the included studies. The time points for lab measurements, radiological evaluation and the disease severity assessment were

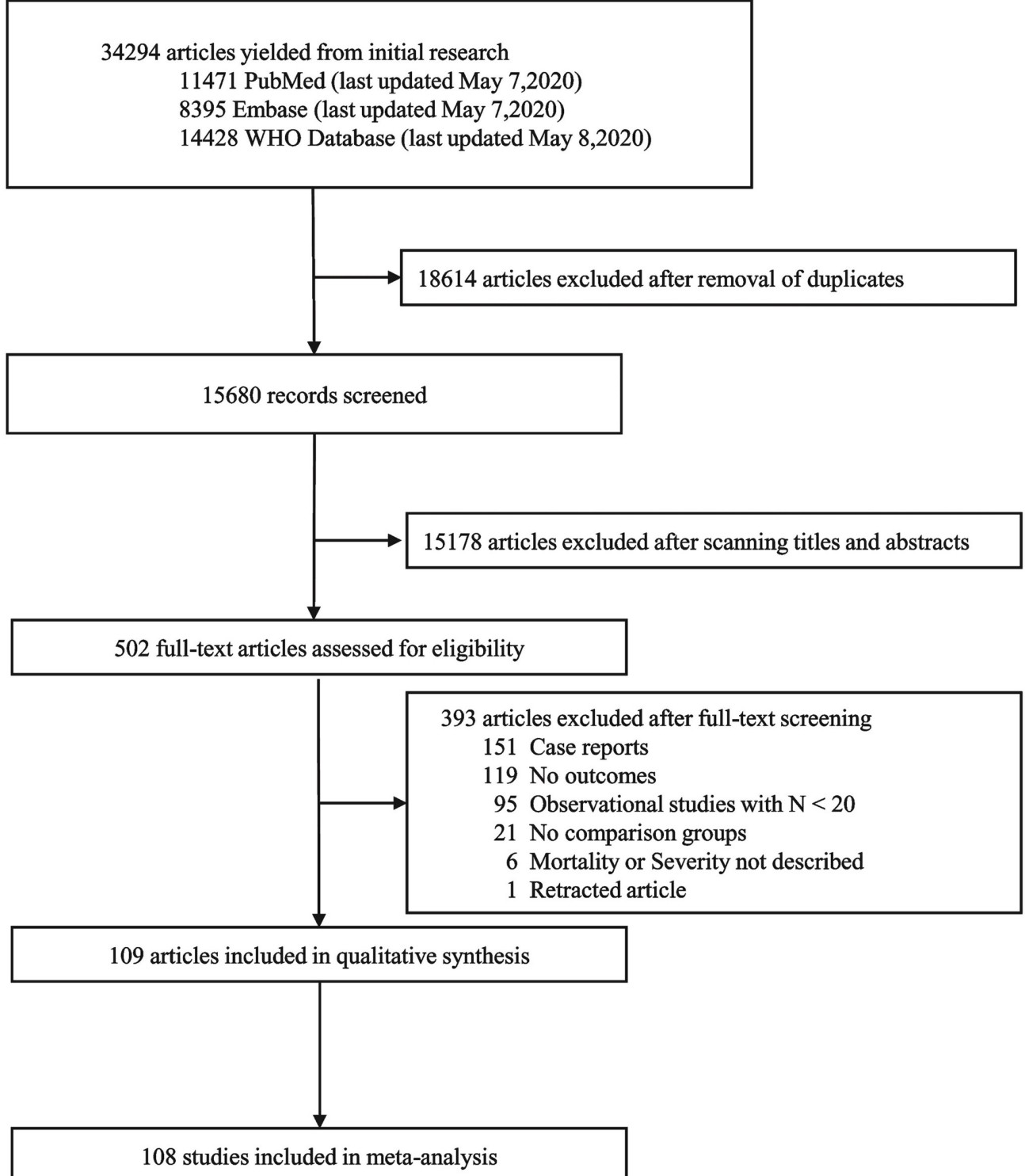

**Fig 1. Stages of the analysis process, assessing inclusion and exclusion criteria that amounted to the articles addressed in this review.**

extracted for the included studies. Continuous data for the age and laboratory variables were documented as mean. The median values reported in studies were transformed into mean [16].

The risk assessment for bias for all the studies included in this review was performed using the Newcastle-Ottawa quality assessment scale (NOS) for observational and cohort studies [17]. The three major domains of quality of a study covered by this tool were selection of participants, comparability of cohorts and outcome assessment against a total score of 9. This was performed independently by at least two of the following authors (RS, MM, ES, SA). When conflicts related to bias arose, the final decision was taken by a consensus between VC and NT.

## Data synthesis and analysis

We performed a meta-analysis with random effects model to obtain pooled effect sizes for the outcomes of interest. The associations between binary parameters and mortality were reported using pooled risk ratios (with 95%CI). Due to the lack of consistency in the time point of assessment of disease severity, odds ratios (with 95%CI) were used to determine the association between the various factors and the presence of severe disease. When the laboratory parameters were reported using different cut-offs, we reported the effect sizes for each cut-off taken separately in addition to the pooled effect sizes for all the cut-offs taken together. Statistical heterogeneity across the studies was assessed by forest plots, $I^2$ and $Tau^2$ statistics. When the $I^2$ was more than 60%, we performed subgroup analyses based on the whether the studies included all patients with COVID-19 or only the patients who were critically ill. If the heterogeneity was still higher than 60%, we performed sensitivity analyses by excluding studies with a low quality (NOS $\leq$ 5). Sensitivity analyses for the laboratory and radiological parameters were also performed by excluding studies which did not report the time points of assessment. Publication bias was assessed by visual inspection of the funnel plot, and Egger's test was performed for those exposures reported by at least 10 studies. For continuous variables, meta-regression was performed to assess the percentage change in mortality or the presence of severe disease with unit increase in the mean of laboratory parameter reported in the studies. For the binary exposures, unadjusted effect sizes were calculated from the summary data. Adjusted effect sizes were not used due to the lack of consistency in the parameters that they are adjusted for, among the included studies. All analyses were carried out using the meta package in Stata (StataCorp, version 16) [18].

## Role of the funding source

This study was not supported by any funding source.

## Results

We identified 15680 studies from three databases after removing duplicates, of which the full text was retrieved for 502 articles. All the articles included were in the English language. The reasons for exclusion of studies are outlined in Fig 1. A total of 109 studies were included for this review, of which 42 studies assessed mortality risks [19–60]; 72 studies determined association with severe disease [24, 34, 39, 41, 56, 58, 61–126], out of which 5 studies reported both the outcomes [24, 34, 39, 56, 58]. Of the total 109 studies in the review, 101 were retrospective cohorts, seven were prospective cohorts, and one of them was an ambispective cohort study. There were no cross-sectional or case control studies that satisfied the inclusion criteria. Of the studies that determined the risk of mortality, an aggregate of 20296 participants were assessed, with 32 studies from China, six from the United States, two from Spain, one from the United Kingdom, one from Italy, one from Iran and a multi-country study. Five of these studies only

included patients who were critically ill or invasively ventilated [19, 23, 32, 35, 48]. Among the studies that assessed the association with severe disease, a total of 17992 participants were included with seventy-one studies from China and one study from Italy. The characteristics of the included studies, along with the time points for laboratory, radiological and disease severity assessment, are outlined in the S1 and S2 Tables (in S1 Appendix Section II).

Quality assessment was performed using the New-Castle Ottawa scale (NOS) as all of the studies used a cohort study design. This revealed that one of the studies (0.9%) had scored 9, 54 studies (49.5%) had scored 8, 39 studies (35.7%) scored 7, six studies (5.5%) scored 6. A total of nine studies were identified as low-quality studies (NOS ≤ 5) with six studies (5.5%) scoring 5, and the remaining three studies (2.7%) scoring 4 (S3 and S4 Tables—in S1 Appendix section III).

Table 1 and Fig 2 show the list of exposures and their relationship with mortality. Males (RR 1.45, 95%CI 1.23–1.71) and ever-smokers (RR 1.43, 95%CI: 1.09–1.87) had higher risk of mortality. Patients with diabetes, hypertension, cardiovascular diseases, chronic renal disease, chronic liver disease and malignancy were associated with an increased risk of mortality; while hepatitis B and HIV infections did not result in higher risk of mortality. Section V of the supplementary document in S1 Appendix illustrates the forest and funnel plots of these associations.

Patients who had dyspnea (RR 2.55, 95%CI: 1.88–3.46) and hemoptysis (RR 1.62, 95%CI: 1.25–2.11) had a significantly higher risk of mortality while other clinical features like fever, sore throat, cough, expectoration, vomiting, diarrhea, nausea, myalgia, headache, anorexia, chest pain, abdominal pain, palpitations, and anosmia did not demonstrate any significant association with mortality. Acute respiratory distress syndrome (ARDS) was associated with an RR of 20.19 [95%CI: 10.87–37.52] with an $I^2$ of 79%. Cardiac complications such as acute cardiac injury and acute cardiac failure had higher risk of mortality with RR of 5.42 [95%CI: 3.79–7.77] and 3.10 [95%CI: 2.55–3.77] respectively. Other complications with higher risk of mortality were sepsis, bacteremia, shock, disseminated intravascular coagulation (DIC), acute kidney injury and acute liver dysfunction. Association of laboratory parameters (based on specific cut-offs) with the risk of mortality are mentioned in the Table 2. Among the laboratory parameters that were assessed using binary cut-offs, increased total leucocyte count, increased neutrophil count, decreased lymphocyte count and reduced platelet count were associated with increased risk of death. Inflammatory parameters such as C-reactive protein and procalcitonin were associated with increased risk of death with RRs of 5.49 [95%CI: 1.72–17.51] and 3.09 [95%CI: 2.35–4.07] respectively. Abnormal renal and liver parameters, hypernatremia and hyperkalemia also were associated with increased risk of mortality.

Presence of bilateral lung infiltrates (RR 1.35, 95%CI 1.07–1.69), consolidation (RR 2.07, 95%CI 1.35–3.16) and air-bronchogram (RR 3.56, 95%CI 1.37–9.29) on CT were associated with increased RR for mortality. The presence of unifocal involvement on CT had lower risk of death (RR 0.31, 95%CI 0.11–0.92). The presence of ground glass opacities or pleural effusion on CT demonstrated no association with mortality.

The odds ratios (OR) of severe disease in patients with various clinical characteristics are shown in Table 1 and Fig 3. The odds of severe disease were high in patients with diabetes, hypertension, cardiovascular diseases, chronic kidney disease, chronic liver disease and chronic obstructive pulmonary disease. HIV and hepatitis B infections were not associated with severe disease. Odds ratios for severe disease were higher in patients with fever, cough, expectoration, anorexia, chest pain, dyspnea, and hemoptysis. The OR for severe disease in patients who had dyspnea was 4.72 [95% CI: 3.18–7.01] among 34 studies. Gastrointestinal symptoms nausea, vomiting and diarrhea did not show association with disease severity.

**Table 1. Association of clinical characteristics with mortality and severe disease in patients with COVID-19.**

| Clinical characteristics | Risk of Mortality | | | | | Odds of Severe disease | | | | |
|---|---|---|---|---|---|---|---|---|---|---|
| | No. of Studies | No. of Patients | Pooled RR [95% CI] | Heterogeneity I² | Heterogeneity T² | No. of Studies | No. of Patients | Pooled OR [95% CI] | Heterogeneity I² | Heterogeneity T² |
| **Baseline characteristics and co-morbidities** | | | | | | | | | | |
| Male sex | 26 | 16422 | *1.45 [1.23–1.71] | 49% | 0.07 | 59 | 17063 | 1.38 [1.24–1.53] | 31% | 0.04 |
| Ever smoker | 7 | 10419 | 1.43 [1.09–1.87] | 0 | 0 | 10 | 4511 | 1.51 [1.06–2.14] | 62% | 0.33 |
| Diabetes | 27 | 16263 | *1.59 [1.41–1.78] | 23% | 0.02 | 36 | 7552 | 2.09 [1.66–2.64] | 40% | 0.16 |
| Hypertension | 26 | 15947 | *1.90 [1.69–2.15] | 28% | 0.02 | 33 | 7002 | 2.63 [2.08–3.33] | 64% | 0.25 |
| Cardiovascular diseases | 25 | 16576 | 2.27 [1.88–2.79] | 71% | 0.13 | 31 | 6932 | 2.83 [2.21–3.63] | 23% | 0.09 |
| Congestive Heart Failure | 5 | 9910 | 2.08 [1.54–2.80] | 0% | 0 | 3 | 558 | 4.76 [1.34–16.97] | 0% | 0 |
| Cerebrovascular Disease | 15 | 2437 | 2.63 [1.97–3.51] | 75% | 0.20 | 13 | 4246 | 2.62 [1.76–3.90] | 7% | 0.04 |
| COPD | 15 | 9717 | 2.29 [1.90–2.75] | 0% | 0 | 19 | 4790 | 3.23 [1.97–5.31] | 24% | 0.27 |
| Asthma | 2 | 643 | 0.98 [0.42–2.32] | 0% | 0 | .. | .. | .. | .. | .. |
| CKD | 15 | 6556 | 2.24 [1.78–2.81] | 39% | 0.07 | 14 | 4442 | 2.62 [1.46–4.71] | 27% | 0.31 |
| Chronic liver disease | 6 | 3672 | 2.18 [1.40–3.40] | 20% | 0.07 | 17 | 8869 | 1.56 [1.12–2.17] | 0% | 0 |
| Hepatitis B infection | 2 | 822 | 1.14 [0.61–2.12] | 0% | 0 | 3 | 1945 | 0.54 [0.17–1.71] | 0% | 0 |
| HIV | 1 | 274 | 1.21 [0.17–8.64] | .. | .. | 2 | 397 | 4.86 [0.50–47.22] | 0% | 0 |
| Cancer | 18 | 7008 | 1.52 [1.21–1.90] | 0% | 0 | 20 | 6026 | 2.90 [1.99–4.24] | 4% | 0.04 |
| Immunodeficiency | .. | .. | .. | .. | .. | 4 | 1838 | 2.51 [0.62–10.10] | 16% | 0.36 |
| Endocrine diseases | .. | .. | .. | .. | .. | 4 | 1378 | 2.45 [1.49–4.04] | 0% | 0 |
| **Clinical features** | | | | | | | | | | |
| Fever | 21 | 3551 | 0.82 [0.67–1.00] | 28% | 0.05 | 37 | 7501 | 1.75 [1.32–2.31] | 56% | 0.31 |
| Sore throat | 4 | 1256 | 0.79 [0.44–1.42] | 0% | 0 | 20 | 4721 | 0.79 [0.60–1.04] | 0% | 0 |
| Cough | 22 | 4098 | 1.00 [0.88–1.14] | 16% | 0.01 | 39 | 7746 | 1.22 [1.08–1.38] | 5% | 0.01 |
| Expectoration | 9 | 1977 | 1.19 [0.98–1.45] | 55% | 0.04 | 21 | 4960 | 1.55 [1.27–1.90] | 32% | 0.06 |
| Vomiting | 5 | 1644 | 0.86 [0.51–1.44] | 0% | 0 | 7 | 1919 | 1.02 [0.65–1.60] | 0% | 0 |
| Diarrhea | 14 | 3230 | 1.15 [0.85–1.57] | 65% | 0.17 | 33 | 6831 | 1.31 [1.00–1.71] | 43% | 0.19 |
| Nausea | 7 | 1725 | 1.00 [0.67–1.50] | 1% | 0 | 13 | 3809 | 1.01 [0.61–1.68] | 49% | 0.34 |
| Myalgia | 18 | 3270 | 0.91 [0.76–1.08] | 8% | 0.01 | 25 | 5831 | 1.22 [0.90–1.65] | 62% | 0.29 |
| Headache | 11 | 2645 | 0.95 [0.66–1.37] | 21% | 0.07 | 26 | 6340 | 1.44 [1.00–2.06] | 53% | 0.37 |
| Anorexia | 5 | 1119 | 1.04 [0.86–1.26] | 0% | 0 | 10 | 1157 | 2.72 [1.84–4.01] | 0% | 0 |
| Chest pain | 7 | 2026 | 1.16 [0.77–1.75] | 43% | 0.11 | 16 | 3558 | 2.70 [1.56–4.68] | 75% | 0.73 |
| Dyspnea | 20 | 3595 | 2.55 [1.88–3.46] | 77% | 0.30 | 34 | 7356 | 4.72 [3.18–7.01] | 86% | 0.90 |
| Hemoptysis | 6 | 1014 | 1.62 [1.25–2.11] | 0% | 0 | 5 | 2584 | 2.93 [1.47–5.83] | 0% | 0 |
| Abdominal pain | 5 | 1498 | 1.22 [0.47–3.16] | 7% | 0.09 | 9 | 2506 | 2.86 [1.00–8.13] | 69% | 1.67 |
| Palpitations | 1 | 225 | 0.94 [0.59–1.49] | .. | .. | 2 | 254 | 3.14 [0.88–11.19] | 0% | 0 |
| Rhinorrhea | 1 | 52 | 1.48 [0.96–2.29] | .. | .. | 4 | 1096 | 0.94 [0.38–2.30] | 0% | 0 |
| Anosmia | 1 | 95 | 0.45 [0.03–6.40] | .. | .. | .. | .. | .. | .. | .. |
| **Complications** | | | | | | | | | | |
| ARDS | 14 | 2795 | *20.19 [10.87–37.52] | 79% | 0.90 | .. | .. | .. | .. | .. |
| Shock | 9 | 1844 | 6.12 [3.59–10.45] | 93% | 0.54 | .. | .. | .. | .. | .. |
| Sepsis | 3 | 573 | 47.95 [11.81–194.72] | 0% | 0 | .. | .. | .. | .. | .. |
| Bacteremia | 6 | 1365 | 5.07 [2.02–12.69] | 94% | 1.18 | .. | .. | .. | .. | .. |
| Acute cardiac injury | 14 | 2860 | 5.42 [3.79–7.77] | 86% | 0.36 | .. | .. | .. | .. | .. |
| Acute heart failure | 3 | 495 | 3.10 [2.55–3.77] | 0% | 0 | .. | .. | .. | .. | .. |

*(Continued)*

Table 1. (Continued)

| Clinical characteristics | Risk of Mortality | | | | | Odds of Severe disease | | | | |
| --- | --- | --- | --- | --- | --- | --- | --- | --- | --- | --- |
| | No. of Studies | No. of Patients | Pooled RR [95% CI] | Heterogeneity | | No. of Studies | No. of Patients | Pooled OR [95% CI] | Heterogeneity | |
| | | | | $I^2$ | $T^2$ | | | | $I^2$ | $T^2$ |
| DIC | 4 | 1394 | 3.41 [2.00–5.81] | 95% | 0.28 | .. | .. | .. | .. | .. |
| GI bleeding | 4 | 1028 | 2.53 [1.42–4.49] | 76% | 0.26 | .. | .. | .. | .. | .. |
| Acute Kidney Injury | 15 | 5331 | *4.65 [3.25–6.65] | 95% | 0.42 | .. | .. | .. | .. | .. |
| Acute liver injury | 10 | 4796 | 2.54 [1.77–3.66] | 93% | 0.28 | .. | .. | .. | .. | .. |
| Hepatic encephalopathy | 1 | 109 | 3.99 [2.71–5.87] | .. | .. | .. | .. | .. | .. | .. |
| Ventilator associated pneumonia | 1 | 52 | 0.51 [0.16–1.62] | .. | .. | .. | .. | .. | .. | .. |
| CT features | | | | | | | | | | |
| Peripheral distribution | 1 | 27 | 0.32 [0.05–2.08] | .. | .. | 4 | 313 | 1.40 [0.36–5.50] | 28% | 0.57 |
| Bilateral involvement | 11 | 2067 | 1.35[1.07–1.69] | 0 | 0 | 19 | 3515 | 4.86 [3.19–7.39] | 47% | 0.28 |
| Consolidation | 3 | 243 | 2.07 [1.35–3.16] | 0 | 0 | 9 | 1084 | 3.01 [1.32–6.88] | 76% | 1.09 |
| GGO | 5 | 706 | 1.41 [0.87–2.28] | 55% | 0.15 | 14 | 2629 | 1.63 [1.22–2.17] | 11% | 0.03 |
| Mixed GGO and consolidation | .. | .. | .. | .. | .. | 3 | 263 | 1.63 [0.69–3.85] | 0 | 0 |
| Air bronchogram | 1 | 27 | 3.56[1.37–9.29] | .. | .. | 4 | 333 | 4.79[1.11–20.61] | 85% | 1.85 |
| Nodular infiltrates | 1 | 27 | 0.41 [0.03–5.41] | .. | 0 | 4 | 356 | 1.03 [0.39–2.73] | 46% | 0.45 |
| Hilar Lymphadenopathy | .. | .. | .. | .. | .. | 5 | 397 | 8.34 [2.57–27.08] | 0% | 0 |
| Tree in bud appearance | .. | .. | .. | .. | .. | 1 | 120 | 4.30 [1.07–17.23] | . | 0 |
| Unifocal involvement | 2 | 463 | 0.31 [0.11–0.92] | 44% | 0.27 | 3 | 1237 | 1.13 [0.46–2.81] | 66% | 0.38 |
| Pleural effusion | 2 | 78 | 1.06 [0.27–4.21] | .. | .. | 8 | 642 | 5.30 [2.74–10.26] | 0% | 0 |
| Pleural thickening | .. | .. | .. | .. | .. | 1 | 52 | 1.86[0.35–9.92] | .. | 0 |
| Inter–lobular septal thickening | .. | .. | .. | .. | .. | 2 | 124 | 2.86 [1.06–7.72] | 0% | 0 |
| Bronchiectasis | .. | .. | .. | .. | .. | 2 | 221 | 5.62 [2.22–14.24] | 0% | 0 |
| Linear infiltrates | .. | .. | .. | .. | .. | 3 | 324 | 3.21[1.00–10.25] | 72% | 0.74 |
| Crazy pavement sign | .. | .. | .. | .. | .. | 5 | 650 | 4.52 [2.08–9.81] | 64% | 0.50 |
| Reticular pattern | .. | .. | .. | .. | .. | 4 | 425 | 5.54 [1.24–24.67] | 72% | 1.61 |

Abbreviations: ARDS = Acute Respiratory Distress Syndrome. CKD = Chronic Kidney Disease. COPD = Chronic Obstructive Pulmonary Disease. CT = Computed Tomography.

DIC = Disseminated Intravascular Coagulation. GGO = Ground Glass Opacity. GI Bleeding = Gastrointestinal Bleeding. HIV = Human Immunodeficiency Virus. OR = Odds ratio. RR = Risk ratio.

*denotes presence of publication bias by Egger's test (p–value < 0.05).

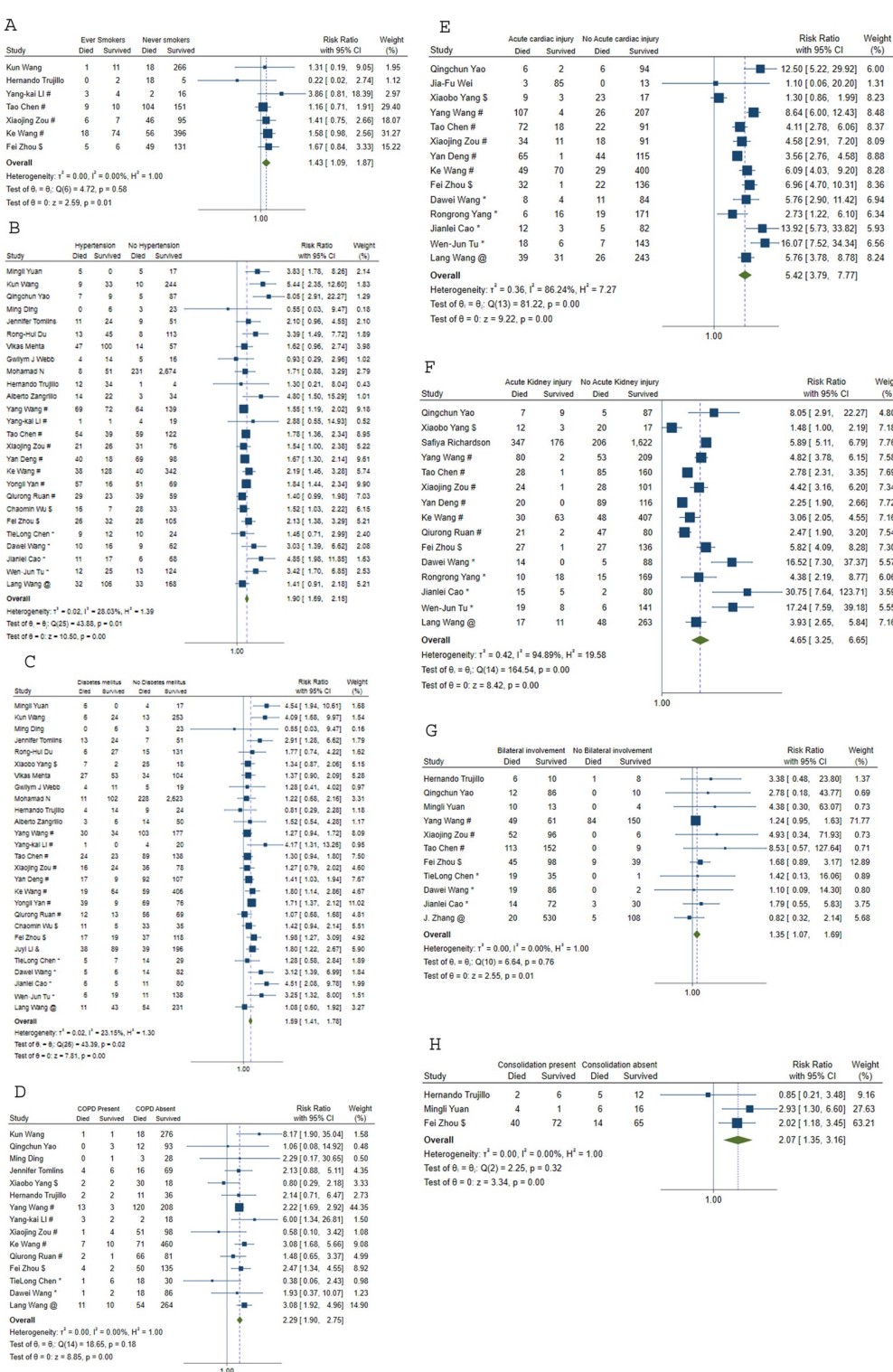

**Fig 2.** Meta-analysis to assess risk of mortality for (A) smoking status. (B) hypertension. (C) diabetes mellitus. (D) COPD. (E) acute cardiac injury. (F) acute kidney injury. (G) bilateral lung involvement. (H) Lung consolidation. COPD = chronic obstructive pulmonary disease. !Shenzhen Third People's Hospital, Shenzhen, China. #Tongji Hospital, Wuhan, China. $Wuhan Jinyintan Hospital, Wuhan, China. *Zhongnan Hospital of Wuhan University, Wuhan, China. %Tianyou Hospital, Wuhan, China. &Taizhou Public Health Medical Center, Zhejiang, China. +Chongqing Three Gorges Central Hospital, Chongqing, China. @Renmin Hospital of Wuhan University, Wuhan, China. ^General Hospital of Central Theater Command of People's Liberation Army, Wuhan, China.

**Table 2. Association of laboratory parameters with mortality in patients with COVID-19.**

| Laboratory Parameters | Pooled Risk of Mortality | | | | | Lab Cut-offs | Risk of Mortality (based on cut-offs) | | | | |
|---|---|---|---|---|---|---|---|---|---|---|---|
| | No. of Studies | No. of Patients | Pooled RR [95% CI] | I² (Heterogeneity) | T² | | No. of Studies | No. of Patients | Pooled OR [95% CI] | I² (Heterogeneity) | T² |
| Decreased TLC | 8 | 1534 | 0.43[0.31–0.59] | 0% | 0 | TLC < 3.5 x10⁹/L | 1 | 154 | 0.58[0.26–1.30] | - | - |
| | | | | | | TLC < 4.0 x10⁹/L | 7 | 1380 | 0.43 [0.29–0.58] | 0% | 0 |
| Increased TLC | 9 | 1708 | 3.85[2.56–5.78] | 80.87% | 0.06 | TLC > 9.5 x10⁹/L | 3 | 353 | 3.60 [2.22–5.84] | 31.88% | 0.06 |
| | | | | | | TLC >10.0 x10⁹/L | 6 | 1355 | 3.97[2.06–7.64] | 90.80% | 0.55 |
| Decreased Neutrophil | 2 | 727 | 0.33[0.03–3.15] | 72.18% | 1.93 | N < 1.8 x10⁹/L | 1 | 179 | 0.94[0.24–3.72] | - | - |
| | | | | | | N < 2 x10⁹/L | 1 | 548 | 0.09[0.01–0.66] | - | - |
| Increased Neutrophil | 3 | 1001 | 4.45 [3.20–6.18] | 38.52% | 0.03 | N > 6.3 x10⁹/L | 2 | 453 | 3.83 [2.90–5.07] | 0% | 0 |
| | | | | | | N > 6.5 x10⁹/L | 1 | 548 | 5.95[3.93–8.99] | - | - |
| Decreased Lymphocyte | 8 | 1554 | 4.09 [1.69–9.91] | 83.63% | 1.24 | L < 0.8 x10⁹/L | 2 | 299 | 4.45[2.70–7.32] | 0% | 0 |
| | | | | | | L < 1.0 x10⁹/L | 2 | 329 | 2.20[0.35–14.07] | 91.96% | 1.65 |
| | | | | | | L < 1.1 x10⁹/L | 4 | 378 | 5.97[0.98–36.29] | 79.77% | 2.68 |
| Decreased Platelet count | 6 | 1439 | 2.42 [1.78–3.30] | 51.04% | 0.07 | Plt count < 100 x 10⁹/L | 3 | 354 | 2.45[1.30–4.62] | 51.44% | 0.16 |
| | | | | | | Plt count < 125 x 10⁹/L | 2 | 537 | 2.45[1.26–4.75] | 74.92% | 0.17 |
| | | | | | | Plt count < 150 x 10⁹/L | 1 | 548 | 2.08 [1.39–3.13] | - | - |
| Decreased Albumin | 4 | 1031 | 3.30[2.61–4.18] | 0% | 0 | Albumin < 32 g/L | 1 | 274 | 3.52[2.61–4.74] | - | - |
| | | | | | | Albumin < 35 g/L | 2 | 702 | 3.27[2.19–4.88] | 0% | 0 |
| | | | | | | Albumin < 40 g/L | 1 | 55 | 1.24[0.36–4.25] | - | - |
| Increased Globulin | 2 | 603 | 1.65[1.15–2.37] | 0% | 0 | Globulin >35 g/L | 2 | 603 | 1.65[1.15–2.37] | 0% | 0 |
| Increased T. Bilirubin | 3 | 810 | 2.74[1.96–3.82] | 0% | 0 | T. Bilirubin > 20 umol/L | 3 | 810 | 2.74[1.96–3.82] | 0% | 0 |
| Increased AST | 5 | 1210 | 2.33[1.93–2.82] | 0% | 0 | AST > 40 U/L | 5 | 1210 | 2.33 [1.93–2.82] | 0% | 0 |
| Increased ALT | 6 | 1330 | 1.48[1.20–1.82] | 8.88% | 0.01 | ALT > 40 U/L | 3 | 976 | 1.30[1.04–1.64] | 0% | 0 |
| | | | | | | ALT > 50 U/L | 3 | 354 | 2.00 [1.39–2.87] | 0% | 0 |
| Increased GGT | ⋮ | ⋮ | ⋮ | ⋮ | ⋮ | ⋮ | ⋮ | ⋮ | ⋮ | ⋮ | ⋮ |
| Increased ALP | ⋮ | ⋮ | ⋮ | ⋮ | ⋮ | ⋮ | ⋮ | ⋮ | ⋮ | ⋮ | ⋮ |
| Increased INR | ⋮ | ⋮ | ⋮ | ⋮ | ⋮ | ⋮ | ⋮ | ⋮ | ⋮ | ⋮ | ⋮ |
| Increased PT | 2 | 345 | 1.97[1.41–2.76] | 0% | 0 | PT > 16s | 2 | 345 | 1.97 [1.41–2.76] | 0% | 0 |
| Elevated CK-total | 4 | 508 | 1.96[1.43–2.70] | 9.52% | 0.01 | CK-total > 171 U/L | 2 | 209 | 1.87[0.88–3.97] | 54.89% | 0.18 |
| | | | | | | CK-total > 185 U/L | 1 | 191 | 1.78[1.09–2.91] | - | - |
| | | | | | | CK-total > 190 U/L | 1 | 108 | 1.56[0.47–5.23] | ⋮ | ⋮ |
| Elevated CK-MB | 1 | 108 | 5.75[2.12–15.62] | - | - | CK-MB > 25 U/L | 1 | 108 | 5.75[2.12–15.62] | ⋮ | ⋮ |
| Increased BUN | 2 | 702 | 4.42[2.99–6.55] | 45.89% | 0.04 | BUN ≥ 7.6 mmol/L | 1 | 548 | 5.34 [3.63–7.85] | ⋮ | ⋮ |
| | | | | | | BUN ≥ 8.3 mmol/L | 1 | 154 | 3.58 [2.32–5.51] | ⋮ | ⋮ |
| Increased Creatinine | 6 | 1235 | 2.92[2.35–3.62] | 0% | 0 | Creatinine > 85 umol/L | 1 | 548 | 2.99[2.35–3.62] | ⋮ | ⋮ |
| | | | | | | Creatinine > 104 umol/L | 1 | 55 | 2.61[1.36–4.98] | ⋮ | ⋮ |
| | | | | | | Creatinine > 115 umol/L | 1 | 154 | 2.88[2.02–4.10] | ⋮ | ⋮ |
| | | | | | | Creatinine > 133 umol/L | 3 | 478 | 3.19 [1.62–6.31] | 42.97% | 0.15 |
| Increased Na⁺ | 2 | 428 | 2.70 [2.00–3.64] | 63% | 0.03 | Na⁺ > 145 mmol/L | 2 | 428 | 2.70 [2.00–3.64] | 63% | 0.03 |
| Increased K⁺ | 3 | 536 | 2.34[1.87–2.94] | 9.45% | 0.00 | K⁺ > 5 mmol/L | 2 | 428 | 2.36[1.85–3.01] | 24.03% | 0.01 |
| | | | | | | K⁺ > 5.4 mmol/L | 1 | 108 | 2.20[0.57–8.54] | 9.45% | 0.00 |
| Decreased K⁺ | 3 | 536 | 0.99[0.43–2.29] | 60.38% | 0.33 | K⁺ < 3.5 mmol/L | 2 | 428 | 0.70[0.22–2.27] | 67.68% | 0.52 |
| | | | | | | K⁺ < 3.8 mmol/L | 1 | 108 | 2.07 [0.69–6.23] | ⋮ | ⋮ |
| Elevated LDH | 5 | 1222 | 5.37 [2.10–13.74] | 80.61% | 0.84 | LDH > 245 U/L | 3 | 345 | 4.19[0.85–20.66] | 83.47% | 1.59 |

(Continued)

**Table 2.** (Continued)

| Laboratory Parameters | Pooled Risk of Mortality — No. of Studies | No. of Patients | Pooled RR [95% CI] | Heterogeneity I² | T² | Lab Cut-offs | Risk of Mortality (based on cut-offs) — No. of Studies | No. of Patients | Pooled OR [95% CI] | Heterogeneity I² | T² |
|---|---|---|---|---|---|---|---|---|---|---|---|
| | | | | | | LDH > 250 U/L | 1 | 548 | 13.10[3.26–52.66] | : | : |
| | | | | | | LDH > 350 U/L | 1 | 274 | 6.33[4.16–9.63] | : | : |
| Elevated Myoglobin | 1 | 179 | 5.30 [2.36–11.92] | : | 0 | Myoglobin> 100 μg/L | 1 | 179 | 5.30 [2.36–11.92] | : | 0 |
| Increased Uric acid | : | : | : | : | : | | : | : | : | : | : |
| Elevated Cystatin C | 1 | 108 | 1.75[0.41-7.50] | : | : | Cystatin C >1.2mg/L | 1 | 108 | 1.75[0.41-7.50] | : | : |
| Elevated D–Dimer | 9 | 2026 | 3.98[2.87–5.52] | 65.13% | 0.13 | D–Dimer > 0.5mg/L | 4 | 562 | 3.33[1.48–7.49] | 63.45% | : |
| | | | | | | D–Dimer > 1mg/L | 3 | 847 | 4.82[3.29–7.07] | 0% | 0 |
| | | | | | | D–Dimer > 2mg/L | 1 | 343 | 5.54[4.15–7.39] | : | : |
| | | | | | | D–Dimer > 21mg/L | 1 | 274 | 3.06[2.44–3.85] | : | : |
| Increased CRP | 6 | 1338 | 5.49[1.72–17.51] | 93.21% | 1.59 | CRP > 10 mg/L | 1 | 108 | 6.22[0.83–46.36] | : | : |
| | | | | | | CRP > 100 mg/L | 5 | 1175 | 5.72[1.40–23.41] | 95.68% | 2.12 |
| Elevated ESR | 2 | 603 | 0.96[0.64–1.44] | 0% | 0 | ESR>15 mm/h | 2 | 603 | 0.96[0.64–1.44] | 0% | 0 |
| Elevated Procalcitonin | 8 | 1555 | 3.09[2.35–4.07] | 53.89% | 0.07 | Procalcitonin >0.05 μg/L | 3 | 810 | 4.45[3.19–6.21] | 0% | 0 |
| | | | | | | Procalcitonin >0.5 μg/L | 4 | 690 | 2.21[1.13–4.33] | 87.85% | 0.35 |
| | | | | | | Procalcitonin >1 μg/L | 1 | 55 | 2.67[1.45–4.90] | : | : |
| Increased Serum ferritin | 1 | 191 | 5.6 [1.47–21.6] | : | : | S. ferritin > 300 μg/L | 1 | 191 | 5.6 [1.47–21.6] | : | : |
| Increased SAA | : | : | : | : | : | | : | : | : | : | : |
| Increased IL–1β | 1 | 274 | 0.84 [0.38–1.83] | : | : | IL–1β > 5 ng/L | 1 | 274 | 0.84 [0.38–1.83] | : | : |
| Increased IL–2 | 1 | 274 | 4.04 [2.18–7.49] | : | : | IL–2 > 710 U/L | 1 | 274 | 4.04 [2.18–7.49] | : | : |
| Increased IL–6 | 2 | 503 | 22.59[3.19–160.03] | 0% | 0 | IL–6 > 2.9 ng/L | 1 | 229 | 12.66[0.79–202.89] | : | : |
| | | | | | | IL–6 > 7 ng/L | 1 | 274 | 40.13[2.53–636.24] | : | : |
| Increased IL–8 | 1 | 274 | 2.29 [1.51–3.45] | : | : | IL–8 > 62 ng/L | 1 | 274 | 2.29 [1.51–3.45] | : | : |
| Increased IL–10 | 1 | 274 | 4.19 [2.56–6.84] | : | : | IL–10 > 9.2 ng/L | 1 | 274 | 4.19 [2.56–6.84] | : | : |
| Increased TNF–Alpha | 1 | 274 | 2.57 [1.46–4.52] | : | : | TNF–Alpha >8.1 ng/L | 1 | 274 | 2.57 [1.46–4.52] | : | : |
| Increased NT–ProBNP | 2 | 822 | 5.48 [3.78–7.34] | 0% | 0 | NT–ProBNP > 285 ng/L | 1 | 275 | 5.87[3.43–10.03] | : | : |
| | | | | | | NT–ProBNP > 500 ng/L | 1 | 548 | 5.136[3.07–8.60] | : | : |
| Elevated troponin | 6 | 1006 | 4.49[3.74–5.38] | 0% | 0 | hs Tn I > 15.6 ng/L | 2 | 375 | 4.62[2.80–7.65] | 52.63% | 0.07 |
| | | | | | | hs Tn I > 26.2 ng/L | 1 | 107 | 6.01[2.80–7.65] | : | : |
| | | | | | | hs Tn I > 28 ng/L | 1 | 191 | 4.29[3.05–6.05] | : | : |
| | | | | | | Tn I > 50 ng/L | 1 | 179 | 5.47[2.44–12.28] | : | : |
| | | | | | | hs Tn I > 100 ng/L | 1 | 154 | 4.28[2.91–6.29] | : | : |

Abbreviations: ALP = Alkaline Phosphatase. ALT = Alanine Aminotransferase. AST = Aspartate Aminotransferase. BNP = Brain Natriuretic Peptide. BUN = Blood Urea Nitrogen. CK = Creatine Kinase. CRP = C–Reactive Protein. ESR = Erythrocyte Sedimentation Rate. GGT = Gamma–Glutamyl Transferase. IL = Interleukin. INR = International Normalized Ratio. K+ = Potassium. LDH = Lactate Dehydrogenase. N = Neutrophil count. Na+ = Sodium. NT–ProBNP = N–Terminal Fragment Brain Natriuretic Peptide. Plt count = Platelet count. OR = Odds ratio. PT = Prothrombin Time. RR = Risk ratio. T Bilirubin = Total Bilirubin. TLC = Total Leucocyte Count. TNF = Tumor Necrosis Factor.

*denotes presence of publication bias by Egger's test (p–value < 0·05).

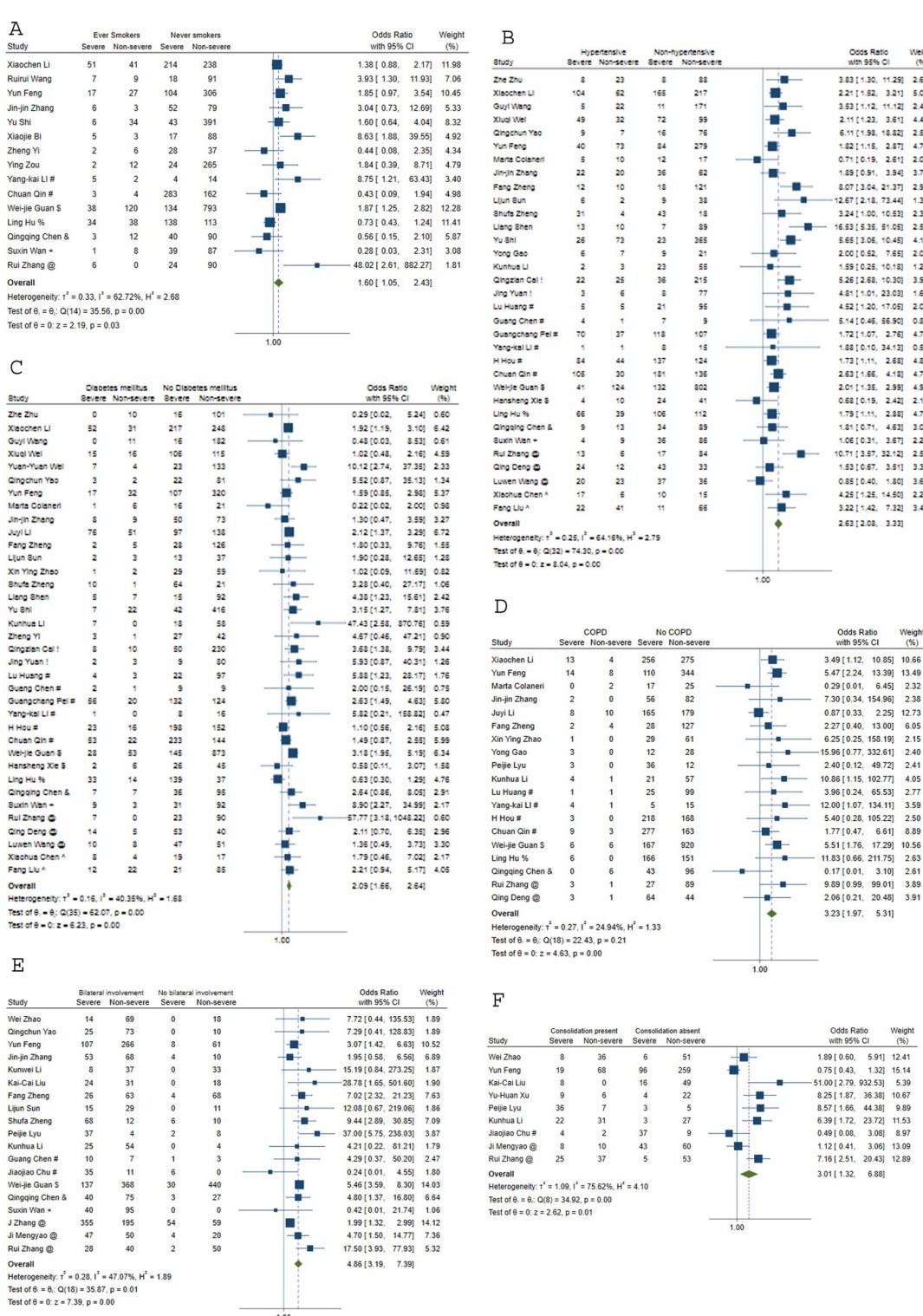

**Fig 3. Meta-analysis to assess odds of severe disease for (A) smoking status (B) hypertension (C) diabetes mellitus (D) COPD E. bilateral lung involvement (F) Lung consolidation.** COPD = chronic obstructive pulmonary disease. !Shenzhen Third People's Hospital, Shenzhen, China. # Tongji Hospital, Wuhan, China. $Wuhan Jinyintan Hospital, Wuhan, China. *Zhongnan Hospital of Wuhan University, Wuhan, China. %Tianyou Hospital, Wuhan, China. &Central hospital of Wuhan, Wuhan. +Chongqing Three Gorges Central Hospital, Chongqing, China. @Renmin Hospital of Wuhan University, Wuhan, China. ^General Hospital of Central Theater Command of People's Liberation Army, Wuhan, China.

Among the laboratory parameters that were assessed for association with severe disease using binary cut-offs (Table 3), decreased lymphocyte count (OR 2.64, 95%CI 1.12–6.23), decreased serum albumin (OR 5.63, 95%CI 1.45–21.87), increased aspartate transaminase (OR 4.33, 95%CI 2.17–8.63) demonstrated increased odds of severe disease. Inflammatory parameters namely, elevated CRP (OR 3.32, 95%CI 1.92–5.71), elevated procalcitonin (OR 5.50, 95% CI 3.38–8.93) and increased ESR (OR 1.68, 95%CI 1.13–2.49) had high ORs for severe disease. Bilateral lung involvement on CT was associated with an OR of 4.86[95%CI: 3.19–7.39] among 19 studies. Other radiological features on CT with higher odds of severe disease were lung consolidation, ground glass opacities, air bronchogram, hilar lymphadenopathy, pleural effusion, crazy-pavement pattern, reticular pattern, tree-in bud appearance, inter-lobar septal thickening, and bronchiectasis. However, the OR of severe disease were not increased with peripheral distribution of infiltrates, nodular infiltrates, linear infiltrates, unifocal involvement or pleural thickening.

Meta-regression of continuous variables (Table 4) revealed that with every ten-year increase in the mean age of the patients, there was a 7.6% and 11.2% increase in the mortality (p-value = 0.02) and disease severity (p-value <0.001) respectively. Increase in the mean total leukocyte count (p-value = 0.04) and a decrease in the mean lymphocyte count (p-value = 0.02) were significantly associated with higher mortality. Higher levels of mean C-reactive protein and mean D-dimer during admission were linked to higher proportion of both severe disease as well as mortality. Increasing mean lactate dehydrogenase and creatine kinase among the included studies were associated with higher mortality but not severe disease. Higher serum creatinine levels and lower serum albumin levels correlated with disease severity but was not significantly associated with mortality, though the direction of the relationship was consistent. Unit increase in the mean blood urea nitrogen resulted in 5.3% increase in the mortality (p-value = 0.030). Our analysis did not reveal a significant association between mortality or disease severity and other laboratory parameters, such as platelet count, hemoglobin, prothrombin time, activated partial thromboplastin time (aPTT), procalcitonin, aspartate transaminase (AST), alanine transaminase (ALT), erythrocyte sedimentation rate, total bilirubin and interleukin-6 levels. Bubble plots assessing the linear relationship between the variables and the outcomes (proportion with severe disease and proportion who died) are shown in Figs 4 and 5 and Supplementary Section VI in S1 Appendix.

Subgroup analyses based on the restriction of inclusion to critically ill participants are shown in S5 Table and in the Supplementary Figures in S1 Appendix Section IV. Subgroup analysis did not significantly reduce heterogeneity between studies except for the association of elevated procalcitonin levels and presence of diarrhea with mortality. The magnitude of the effect size changed significantly in the subgroups however the direction of the effect remained consistent for multiple parameters, including dyspnea, cardiovascular disease, cerebrovascular disease, diabetes mellitus, bacteremia, and gastrointestinal bleeding. In studies with critically ill patients, the direction of the effect was reversed for gender and elevated procalcitonin levels, but the association was not statistically significant in the subgroups. Sensitivity analysis by excluding low-quality studies (NOS≤5) did not significantly reduce heterogeneity or alter the pooled effect sizes for the exposures with $I^2$>50% for mortality. Sensitivity analysis by excluding low-quality studies reporting COVID-19 severity yielded a decrease in the heterogeneity for various exposures, such as smoking, hypertension, myalgia, and air-bronchogram in CT. Nevertheless, heterogeneity was substantially high for exposures, such as dyspnea, chest pain, abdominal pain, unifocal involvement in CT and certain laboratory parameters. Sensitivity analysis for the laboratory and radiological variables by excluding studies with an unclear timepoint for the assessment of the parameters did not result in change in the direction or significance of the association. In our analysis, bias due to small study effect could not be ruled

**Table 3. Association of laboratory parameters with disease severity in patients with COVID-19.**

| Laboratory Parameters | Pooled Odds of Disease Severity | | | | | Lab Cut-offs | Pooled Odds of Disease Severity (based on cut-offs) | | | | |
|---|---|---|---|---|---|---|---|---|---|---|---|
| | No. of Studies | No. of Patients | Pooled RR [95% CI] | Heterogeneity I² | T² | | No. of Studies | No. of Patients | Pooled OR [95% CI] | Heterogeneity I² | T² |
| Decreased TLC | 11 | 3074 | 0.84[0.52–1.37] | 78.58% | 0.46 | TLC < 3.5 x10⁹/L | 4 | 548 | 0.70[0.37–1.33] | 29.06% | .. |
| | | | | | | TLC < 4.0 x10⁹/L | 6 | 2482 | 0.89[0.43–1.82] | 88.86% | 0.67 |
| | | | | | | TLC < 5.0 x10⁹/L | 1 | 44 | 1.21[0.36–4.08] | .. | .. |
| Increased TLC | 11 | 3353 | 3.70[2.31–5.93] | 49.71% | 0.27 | TLC > 9.5 x10⁹/L | 4 | 548 | 2.94[2.31–5.93] | 47.12% | 0.47 |
| | | | | | | TLC >10.0 x10⁹/L | 6 | 1706 | 4.65[2.17–9.98] | 63.09% | 0.49 |
| | | | | | | TLC >11.0 x10⁹/L | 1 | 1099 | 2.54[1.43–4.52] | .. | .. |
| Decreased Neutrophil | 2 | 689 | 0.33[0.19–0.58] | 0% | 0 | N < 1.8 x10⁹/L | 1 | 141 | 0.95[0.10–8.97] | .. | .. |
| | | | | | | N < 2 x10⁹/L | 1 | 548 | 0.31[0.17–0.55] | .. | .. |
| Increased Neutrophil | 3 | 554 | 2.00[0.93–4.31] | 52.14% | 0.25 | N > 6.3 x10⁹/L | 1 | 141 | 4.95[1.67–14.68] | .. | .. |
| | | | | | | N > 6.5 x10⁹/L | 1 | 90 | 1.19[0.32–4.42] | .. | .. |
| | | | | | | N > 7.5 x10⁹/L | 1 | 323 | 1.52[0.93–2.47] | .. | .. |
| Decreased Lymphocyte | 7 | 2922 | 2.64[1.12–6.23] | 87.02% | 1.04 | L < 0.8 x10⁹/L | 2 | 342 | 3.10[0.49–19.69] | 83.84% | 1.49 |
| | | | | | | L < 1.0 x10⁹/L | 1 | 476 | 4.04[2.59–6.31] | .. | .. |
| | | | | | | L < 1.1 x10⁹/L | 2 | 638 | 1.77[0.09–33.23] | 94.51% | 4.23 |
| | | | | | | L < 1.5 x10⁹/L | 1 | 1143 | 8.56[0.44–165.38] | .. | .. |
| | | | | | | L < 2.0 x10⁹/L | 1 | 323 | 1.84[1.12–6.23] | .. | .. |
| Decreased Platelet count | 9 | 2691 | 2.39[1.72–3.34] | 34.31% | 0.13 | Plt count < 100 x 10⁹/L | 3 | 592 | 2.27[1.08–4.78] | 0% | 0 |
| | | | | | | Plt count < 125 x 10⁹/L | 3 | 408 | 3.09[1.37–6.99] | 45.21% | 0.23 |
| | | | | | | Plt count < 150 x 10⁹/L | 3 | 1691 | 2.22[1.32–3.73] | 67.23% | 0.13 |
| Decreased Albumin | 2 | 663 | 5.63[1.45–21.87] | 77.32% | 0.77 | Albumin < 35 g/L | 1 | 548 | 3.19[2.22–4.57] | .. | .. |
| | | | | | | Albumin < 40 g/L | 1 | 115 | 13.07[3.68–46.40] | .. | .. |
| Increased Globulin | 2 | 663 | 1.98[1.44–2.72] | 0% | 0 | Globulin >30 g/L | 1 | 115 | 1.67[0.77–3.64] | 0% | 0 |
| | | | | | | Globulin >35 g/L | 1 | 548 | 2.05[1.44–2.90] | 0% | 0 |
| Increased T. Bilirubin | 5 | 2031 | 2.23[1.34–3.71] | 23.14% | 0.08 | T. Bilirubin > 20 umol/L | 5 | 2031 | 2.23[1.34–3.71] | 23.14% | 0.08 |
| Increased AST | 7 | 2624 | 4.85[2.52–9.34] | 86.47% | 0.62 | AST > 40 U/L | 7 | 2624 | 4.85[2.52–9.34] | 86.47% | 0.62 |
| Increased ALT | 7 | 2920 | 2.40[1.09–5.29] | 89.03% | 0.97 | ALT > 40 U/L | 4 | 2607 | 2.36[0.81–6.89] | 97.33% | 2.41 |
| | | | | | | ALT > 50 U/L | 3 | 313 | 2.33[0.85–6.38] | 0% | 0 |
| Increased GGT | 1 | 115 | 1.42[0.44–4.55] | .. | .. | GGT > 57 U/L | 1 | 115 | 1.42[0.44–4.55] | .. | .. |
| Increased ALP | 1 | 115 | 6.07[1.05–35.03] | .. | .. | ALP > 120 U/L | 1 | 115 | 6.07[1.05–35.03] | .. | .. |
| Increased INR | 1 | 115 | 1.38[0.60–3.18] | .. | .. | INR > 1.15 | 1 | 115 | 1.38[0.60–3.18] | .. | .. |
| Increased PT | 2 | 413 | 2.35[1.26–4.41] | 0% | 0 | PT > 12.8s | 1 | 90 | 1.65[0.58–4.67] | .. | .. |
| | | | | | | PT > 14s | 1 | 323 | 2.88[1.31–6.32] | .. | .. |
| Elevated CK-total | 6 | 1733 | 3.11[1.74–5.55] | 41.86% | 0.21 | CK-total > 190 U/L | 3 | 359 | 3.73[1.97–7.07] | 2.42% | 0.01 |
| | | | | | | CK-total > 190 U/L | 3 | 1374 | 2.87[1.03–8.00] | 49.25% | 0.42 |
| Elevated CK-MB | 4 | 794 | 1.41[0.42–4.67] | 72.23% | 1.02 | CK-MB > 5 U/L | 2 | 596 | 0.73[0.14–3.69] | 77.79% | 1.08 |
| | | | | | | CK-MB > 25 U/L | 2 | 198 | 2.83[1.03–7.82] | 0% | 0 |
| Increased BUN | 2 | 871 | 4.06[2.71–6.07] | 0% | 0 | BUN ≥ 7.6 mmol/L | 1 | 548 | 4.77[2.75–8.29] | .. | .. |
| | | | | | | BUN ≥ 8 mmol/L | 1 | 323 | 3.37[1.87–6.09] | .. | .. |
| Increased Creatinine | 8 | 2508 | 2.49[1.41–4.41] | 21.59% | 0.14 | Creatinine > 87 umol/L | 2 | 709 | 1.66[1.14–2.43] | 0% | 0 |
| | | | | | | Creatinine > 97 umol/L | 2 | 225 | 5.88[0.63–54.73] | 47.51% | 1.33 |
| | | | | | | Creatinine > 133 umol/L | 3 | 1251 | 3.81[1.4–10.11] | 0% | 0 |
| | | | | | | Creatinine > 144 umol/L | 1 | 323 | 2.18[0.42–11.41] | .. | .. |

(*Continued*)

Table 3. (Continued)

| Laboratory Parameters | Pooled Odds of Disease Severity | | | | | Lab Cut-offs | Pooled Odds of Disease Severity (based on cut-offs) | | | | |
|---|---|---|---|---|---|---|---|---|---|---|---|
| | No. of Studies | No. of Patients | Pooled RR [95% CI] | Heterogeneity I² | Heterogeneity T² | | No. of Studies | No. of Patients | Pooled OR [95% CI] | Heterogeneity I² | Heterogeneity T² |
| Increased Na⁺ | : | : | : | : | : | | : | : | : | : | : |
| Increased K⁺ | 1 | 108 | 0.94[0.18–4.86] | : | : | K⁺ > 5.1 mmol/L | 1 | 108 | 0.94[0.18–4.86] | : | : |
| Decreased K⁺ | 1 | 108 | 7.59 [2.67–21.60] | : | : | K⁺ < 3.8 mmol/L | 1 | 108 | 7.59 [2.67–21.60] | : | : |
| Elevated LDH | 7 | 2425 | 3.77[1.95–7.30] | 82.69% | 0.61 | LDH > 225 U/L | 1 | 161 | 4.70[2.02–10.94] | : | : |
| | | | | | | LDH > 243 U/L | 1 | 115 | 5.90[2.33–14.95] | : | : |
| | | | | | | LDH > 250 U/L | 4 | 2105 | 2.92[0.95–8.93] | 92.70% | 1.18 |
| | | | | | | LDH > 300 U/L | 1 | 44 | 6.29[1.60–24.73] | : | : |
| Elevated Myoglobin | 1 | 273 | 4.57[2.05–10.17] | : | : | Myoglobin> 100 µg/L | 1 | 273 | 4.57[2.05–10.17] | : | : |
| Increased Uric acid | 1 | 90 | 3.28[0.52–20.77] | : | : | Uric acid >417 umol/L | 1 | 90 | 3.28[0.52–20.77] | : | : |
| Increased Cystatin C | 1 | 108 | 2.14[0.66–6.88] | : | : | Cystatin C >1.2mg/L | 1 | 108 | 2.14[0.66–6.88] | : | : |
| Elevated D–Dimer | 5 | 1985 | 2.75[1.92–3.93] | 32.40% | 0.05 | D–Dimer > 0.25mg/L | 2 | 230 | 3.44[1.48–8.02] | 0% | 0 |
| | | | | | | D–Dimer > 0.5mg/L | 1 | 1099 | 1.94[1.27–2.97] | : | : |
| | | | | | | D–Dimer > 1mg/L | 2 | 656 | 3.33[2.37–4.69] | 0% | 0 |
| Increased CRP | 12 | 3375 | 3.32[1.92–5.71] | 73.74% | 0.55 | CRP > 30 mg/L | 3 | 595 | 1.82[0.63–5.24] | 50.77% | 0.45 |
| | | | | | | CRP > 50 mg/L | 1 | 108 | 9.25[2.05–41.80] | : | : |
| | | | | | | CRP > 80 mg/L | 2 | 301 | 14.43[3.89–53.50] | 0% | 0 |
| | | | | | | CRP > 100 mg/L | 6 | 2371 | 2.92[1.55–5.51] | 77.52% | 0.43 |
| Elevated ESR | 1 | 548 | 1.68[1.13–2.49] | : | : | ESR>20 mm/h | 1 | 548 | 1.68[1.13–2.49] | : | : |
| Elevated Procalcitonin | 8 | 2392 | 5.50[3.38–8.93] | 38.70% | 0.18 | Procalcitonin >0.05 µg/L | 2 | 240 | 3.45[1.61–7.37] | 0% | 0 |
| | | | | | | Procalcitonin >0.1 µg/L | 2 | 275 | 6.34[1.58–25.38] | 78.13% | 0.79 |
| | | | | | | Procalcitonin >0.5 µg/L | 4 | 1877 | 6.97[3.12–15.58] | 39.61% | 0.26 |
| Increased Serum ferritin | 1 | 548 | 3.57 [2.12–6.01] | : | : | S. ferritin > 500 µg/L | 1 | 548 | 3.57 [2.12–6.01] | : | : |
| Increased SAA | 5 | 830 | 1.54[0.84–2.82] | 19.73% | 0.10 | SAA > 1mg/L | 1 | 121 | 0.92[0.43–2.01] | : | : |
| | | | | | | SAA > 10mg/L | 4 | 709 | 2.39[0.92–6.22] | 27.08% | 0.28 |
| Increased IL–1β | 1 | 548 | 0.62[0.37–1.02] | : | : | IL–1β > 5 ng/L | 1 | 548 | 0.62[0.37–1.02] | : | : |
| Increased IL–2 | 1 | 548 | 2.60[1.63–4.14] | : | : | IL–2 receptor > 710 U/L | 1 | 548 | 2.60[1.63–4.14] | : | : |
| Increased IL–6 | 3 | 778 | 3.30[0.73–14.93] | 82.0% | 1.38 | IL–6 > 7 ng/L | 3 | 778 | 3.30[0.73–14.93] | 82.0% | 1.38 |
| Increased IL–8 | 1 | 548 | 1.83[0.79–4.27] | : | : | IL–8 > 62 ng/L | 1 | 548 | 1.83[0.79–4.27] | : | : |
| Increased IL–10 | 1 | 548 | 1.62[0.98–2.67] | : | : | IL–10 >9.2 ng/L | 1 | 548 | 1.62[0.98–2.67] | : | : |
| Increased TNF–Alpha | 1 | 548 | 1.90[1.20–3.03] | : | : | TNF–Alpha >8.1 ng/L | 1 | 548 | 1.90[1.20–3.03] | : | : |
| Increased NT–Pro BNP | 2 | 821 | 4.43[2.80–7.02] | 0% | 0 | NT–Pro BNP > 500 ng/L | 1 | 548 | 4.23[2.36–7.59] | : | : |
| | | | | | | NT–Pro BNP > 900 ng/L | 1 | 283 | 4.78[2.27–10.08] | : | : |
| Elevated troponin | 2 | 596 | 3.04 [1.03–8.97] | 77.67% | 0.48 | hs Tn I > 40 ng/L | 2 | 596 | 3.04 [1.03–8.97] | 77.67% | 0.48 |

Abbreviations: ALP = Alkaline Phosphatase. ALT = Alanine Aminotransferase. AST = Aspartate Aminotransferase. BNP = Brain Natriuretic Peptide. BUN = Blood Urea Nitrogen. CK = Creatine Kinase. CRP = C–Reactive Protein. ESR = Erythrocyte Sedimentation Rate. GGT = Gamma–Glutamyl Transferase. IL = Interleukin. INR = International Normalized Ratio. K+ = Potassium. LDH = Lactate Dehydrogenase. N = Neutrophil count. Na+ = Sodium. NT–ProBNP = N–Terminal Fragment Brain Natriuretic Peptide. Plt count = Platelet count. OR = Odds ratio. PT = Prothrombin Time. RR = Risk ratio. T Bilirubin = Total Bilirubin. TLC = Total Leucocyte Count. TNF = Tumor Necrosis Factor.

* denotes presence of publication bias by Egger's test (p-value < 0·05).

**Table 4. Association of clinical parameters expressed as continuous variables with mortality and disease severity in patients with COVID–19 by meta-regression.**

| Variable | Mean increase of the variable | No· of studies | No· of patients | Percentage change in mortality [95% CI] | p–value | No· of studies | No· of patients | Percentage change in severe disease [95% CI] | p–value |
|---|---|---|---|---|---|---|---|---|---|
| Age | 10 years | 41 | 20296 | 7.6 [1.0, 14.2] | 0.02 | 70 | 17799 | 11.3 [6.5, 16.2] | <0.001 |
| Hb | 1g/L | 12 | 2519 | 1.1 [–1.4, 3.6] | 0.41 | 13 | 3013 | –1.1 [–1.6, –0.5] | 0.002 |
| TLC | 1x10^9/L | 20 | 9797 | 5.4 [0.2, 10.6] | 0.04 | 28 | 5370 | 15.2 [6.23, 24.18] | <0.001 |
| Lymphocyte count | 0·1x10^9/L | 24 | 10097 | –4.1 [–7.4, –0.8] | 0.02 | 31 | 5696 | –17.3 [– 35.9, 1.3] | 0.07 |
| Neutrophil count | 1x10^9/L | 14 | 8930 | 10.5 [0.5, 20.6] | 0.04 | 22 | 3586 | 9.8 [0.5, 19.1] | 0.04 |
| Platelet count | 50 x 10^9/L | 14 | 3120 | –18.8 [–41.6, 3.85] | 0.10 | 15 | 3190 | 0.1 [–0.4, 0.51] | 0.82 |
| Prothrombin time | 1 s | 16 | 3277 | 3.3 [–4.2, 10.8] | 0.39 | ·· | ·· | ·· | ·· |
| aPTT | 1 s | 12 | 2651 | 1.0 [–0.9, 3.0] | 0.24 | ·· | ·· | ·· | ·· |
| CRP | 100 mg/L | 19 | 3974 | 6.1 [3.3, 8.9] | <0.001 | 27 | 4517 | 5.0 [2.9, 7.0] | <0.001 |
| D–dimer | 1mg/L | 15 | 3621 | 21.6 [7.0, 36.4] | <0.01 | 21 | 3387 | 2.9 [0.1, 5.8] | 0.04 |
| Pro–calcitonin | 0·1 mg/L | 10 | 2349 | 4.5 [–19.3, 28.5] | 0.71 | 19 | 2725 | 8.5 [–1.5, 18.4] | 0.09 |
| Serum LDH | 100 U/L | 13 | 2682 | 9.1 [2.4, 15.8] | 0.01 | 12 | 1666 | –0.02 [–0.1, 0.1] | 0.52 |
| Serum creatinine | 10 μmol/L | 19 | 3817 | 0.2 [–0.4, 0.7] | 0.51 | 18 | 3062 | 9 [0.7, 17.5] | 0.03 |
| BUN | 1 mmol/L | 14 | 2979 | 5.3 [0.5, 10.2] | 0.03 | ·· | ·· | ·· | ·· |
| Total Bilirubin | 1 μmol/L | 12 | 2191 | 2.6 [–0.2, 5.5] | 0.07 | 13 | 8040 | –2.0 [–6.8, 2.7] | 0.41 |
| Albumin | 10 g/L | 12 | 1986 | –4.1 [––22.9, 14.6] | 0.67 | 12 | 7789 | –5.0 [–7.2, –2.8] | <0.001 |
| AST | 10 U/L | 15 | 8927 | 1.7 [–0.9, 4.4] | 0.19 | 18 | 8395 | –1.9 [–14.6, 10.8] | 0.77 |
| ALT | 10 U/L | 19 | 9664 | 2.6 [–1.2, 6.4] | 0.18 | 20 | 8836 | 7.4 [– 7.6, 22.5] | 0.33 |
| CK–total | 10 U/L | 10 | 2051 | 1.6 [0.02, 3.1] | 0.04 | 12 | 2331 | –0.3 [–0.7, 0.2] | 0.31 |
| IL–6 | 10 ng/L | ·· | ·· | ·· | ·· | 13 | 2404 | 1 [–0.2, 3] | 0.09 |

Abbreviations: ALT = Alanine Aminotransferase. aPTT = activated Partial Thromboplastin Time. AST = Aspartate Aminotransferase. BUN = Blood Urea Nitrogen. CK = Creatinine Kinase. CRP = C–Reactive Protein. Hb = Hemoglobin. IL = Interleukin. LDH = Lactate Dehydrogenase. TLC = Total Leucocyte Count.

out while assessing risk of mortality for the following exposures, such as gender, diabetes mellitus, hypertension, ARDS and acute kidney injury and hence the results for these factors should be interpreted with caution.

## Discussion

A total of 109 articles were deemed suitable for data synthesis and identification of variables associated with severe COVID-19 disease and mortality. Specific determinants were identified from a array of clinical parameters such as symptoms, co-morbidities, laboratory, and radiological data. Our findings have potential implications for clinical decision-making, as well as allocation of scarce critical care resources for patients with COVID-19.

The presence of various comorbidities was reported to be associated with severe disease and/or death in patients with COVID-19 in prior studies [127]. Although the direction of association was consistent with previous reports, the risks of death in patients with diabetes and hypertension were lower in our study with an RR of 1.59 [95% CI: 1.41–1.78] and 1.90 [95% CI: 1.69–2.15] respectively for mortality. The levels of control of diabetes and hypertension in these patients, as well as pharmacotherapy for these conditions were not taken into consideration in our review, which might account for the clinical heterogeneity. There was a significant association between pre-existing cardiovascular diseases and COVID-19 attributable mortality, with an RR of 2.27 [95%CI: 1.88–2.79]. This is similar to the association of cardiovascular

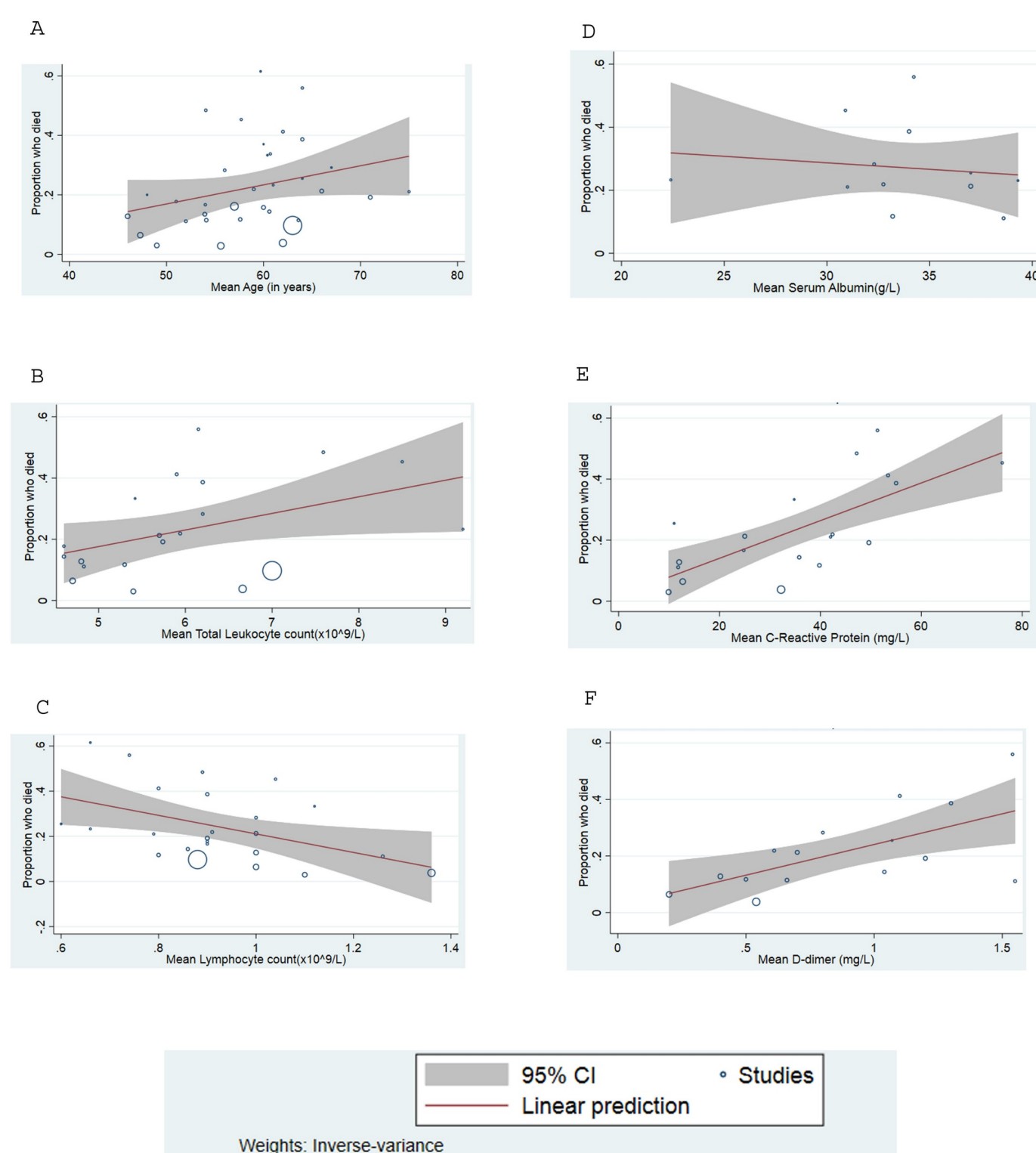

**Fig 4.** Meta-regression plot showing the proportion increase in mortality among COVID-19 patients regressed against (A) mean age. (B) mean leukocyte count. (C) mean lymphocyte count. (D) mean serum albumin. (E) mean C-reactive protein. (F) mean D-dimer.

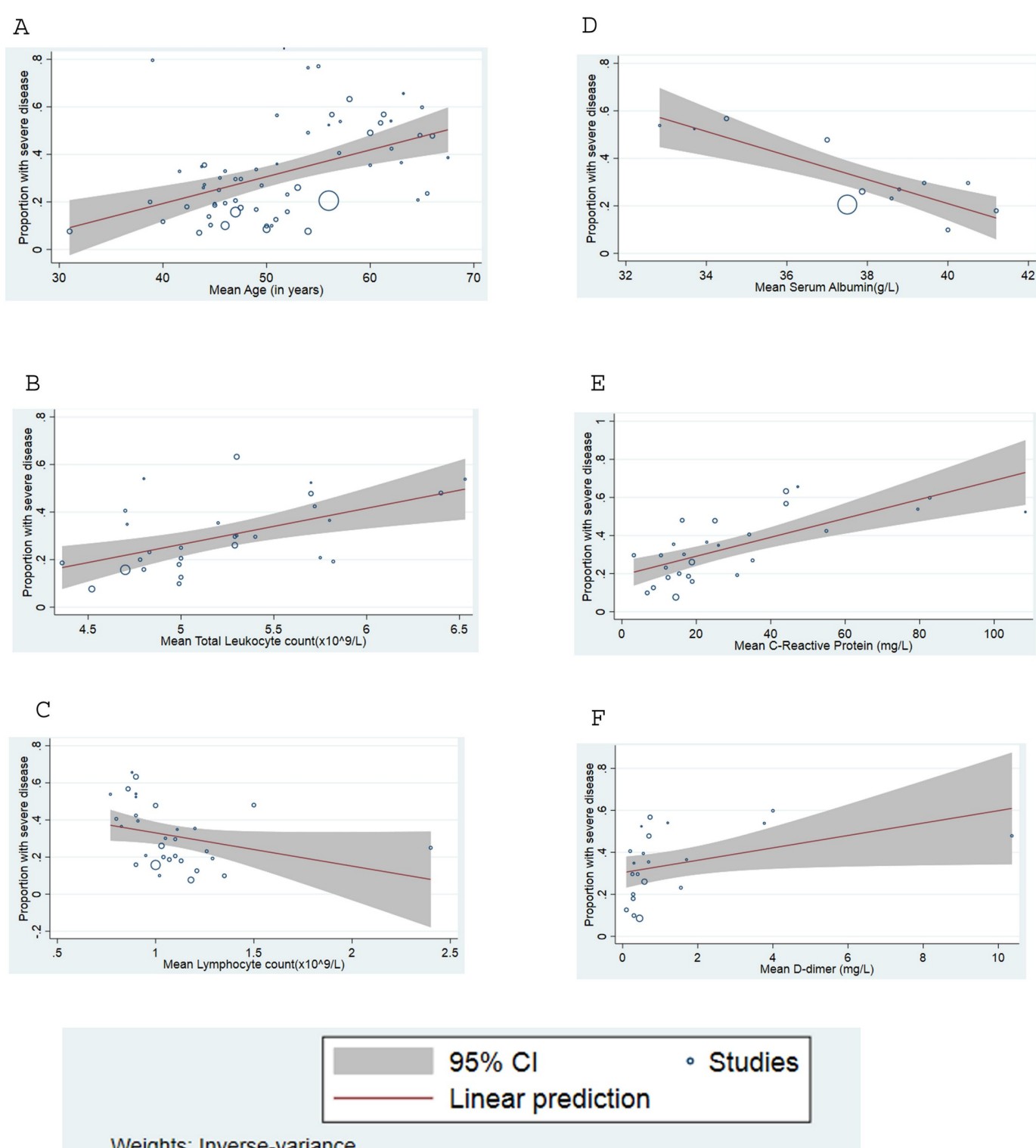

**Fig 5.** Meta-regression plot shows the proportion increase in severity among COVID-19 patients regressed against (A) mean age. (B) mean leukocyte count. (C) mean lymphocyte count. (D) mean serum albumin. (E) mean C-reactive protein. (F) mean D-dimer.

diseases with mortality seen in patients with other viral infections [128, 129]. Chronic kidney disease and chronic liver disease were also associated with higher mortality, but because of the lack of data in the available studies, distinction could not be made with respect to the stage of the kidney and liver dysfunction. Pre-existing diseases of the lung were also associated with adverse outcomes in our study.

The identification of COVID-19-related symptoms associated with mortality and severe disease is especially important since this is among the most readily accessible information during the initial evaluation of patients. Our finding that dyspnea was associated with higher RR of mortality and higher OR for severe disease is consistent with data reported in other retrospective studies on ICU admission and the development of ARDS in patients with COVID-19 [23, 130]. Despite the common occurrence of gastrointestinal symptoms (nausea, vomiting and diarrhea) in patients with COVID-19, no association was found between the presence of these symptoms and the presence of severe disease or mortality in our study.

We found that ARDS had a RR of 20.19 [95%CI: 10.87–37.52] for mortality, which is consistent with a previous study reporting a 28-day survival of 50% among COVID-19 patients with severe ARDS [20]. In contrast to the relatively transient cardiac involvement in SARS-CoV infection [131], we found that cardiac complications such as acute heart failure and acute cardiac injury were associated with a high RR of death in COVID-19 in our study. While the presence of underlying cardiovascular disease increases the risk of developing cardiac complications, Chen *et al* reported that COVID-19 related cardiac complications were also frequent among those without pre-existing cardiovascular diseases [20]. A previous report from 2009 suggested direct cardiac muscle damage in patients with SARS-CoV infection [132]. The cardiac injury seen in patients with COVID-19, might be due to a similar mechanism. Elevated troponin levels were associated with an OR of 3.04 [95% CI: 1.03–8.97] for severe disease in our study. Consistent with our findings, patients with underlying cardiovascular diseases and non-elevated levels of troponin were shown to have lower death rates compared to those without cardiovascular disease but with elevated troponin [133].

Patients with leukocytosis and lymphopenia had higher OR for severe disease and greater RR for mortality. The occurrence of lymphopenia in severe disease may be due to apoptosis of lymphocytes as a result of increased levels of cytokines in the blood in patients with severe disease [134–136]. Hypoalbuminemia was associated with an increased RR of mortality (RR 3.30, 95%CI 2.61–4.18), and is likely related to the systemic inflammatory response in severe COVID-19 [74]. Increased levels of acute phase reactants, such as CRP and ferritin in patients with severe disease, also documented in our review, further supports the inflammatory nature of the disease [85]. Elevated levels of procalcitonin may indicate a secondary bacterial sepsis in patients with severe COVID-19, which was associated with a high RR of mortality in our meta-analysis [137]. COVID-19 has been hypothesized to be a prothrombotic state due to endothelial dysfunction and increased hypoxia-inducible transcription factor in patients with severe pneumonia, and plasminogen activation inhibition in patients who develop sepsis [138–140]. DIC, which is common in patients succumbing to COVID-19, is typically accompanied by elevated D-dimer levels [21]. The levels of D-dimer were found in our meta-analysis to be significantly associated with mortality (RR 3.98, 95%CI 2.87–5.52) and severe disease (OR 2.75, 95%CI 1.92–3.93). Although their association with mortality has not been fully investigated, thromboembolic complications, such as pulmonary embolism and acute stroke, have been noted with increasing frequency in patients with COVID-19 [141, 142].

COVID-19 can manifest in a variety of radiographic patterns on chest CT scan, most of which are consistent with viral pneumonia. We found that bilateral lung involvement and consolidation were associated with higher RR of mortality and higher OR of severe disease. Consistent with our findings, the occurrence of bilateral lung involvement was shown to increase with

disease progression, and is more commonly observed in the late phases of COVID-19 [143]. The ground-glass opacities seen in a large proportion of COVID-19 patients may be due to the thickening of the alveolar septa following inflammation or the incomplete filling of the alveoli, as seen in Influenza A [144]. As noted for other respiratory viruses, our data revealed that consolidation was associated with a higher RR of mortality and OR of severe disease compared to ground glass opacities in patients with COVID-19 [145, 146]. Hilar lymphadenopathy, though rare in COVID-19 patients, could be due to infiltration of the hilar lymph nodes by lymphocytes and macrophages, and appears to be associated with severe disease in our meta-analysis [147–149].

Although this systematic review is able to delineate important parameters associated with disease severity and mortality in COVID-19, our study has a few limitations. First, we included current published articles related to the highly dynamic information available on COVID-19. As this pandemic has not impacted all regions within the same time frame, there is a potential timing bias, whereby the majority of patients described are from early-hit regions, which may not be representative of other patient populations, based on sociodemographic characteristics and pre-existing conditions. Although a pooled analysis of effect sizes adjusted for potential confounders would have been desirable, most studies did not uniformly report estimates adjusted for the same parameters. Time points of evaluation of laboratory, radiological and disease severity assessment were not clearly defined in many of the studies, which precluded the calculation of risk ratio of severe disease. We did not take into account the critical care interventions and strategies that potentially impact the course of the disease and/or survival, as such interventions (e.g. mechanical ventilation, extracorporeal membrane oxygenation, convalescent plasma) vary across different regions. Further investigation is warranted to evaluate which interventions impact the morbidity and mortality of patients with COVID-19. Future studies may investigate if individual and contextual-level sociodemographic factors are associated with morbidity and survival.

In summary, this study comprehensively examined the effect of several demographic, clinical, laboratory and radiological risk factors associated with mortality and severe disease among patients with COVID-19. Knowledge of these risk factors may help health care professionals to develop improved clinical management plans based on risk stratification. Policy makers can utilize these findings to develop triage protocols and effectively allocate resources in resource-limited settings.

## Supporting information

**S1 Appendix.**
(PDF)

**S1 PRISMA Checklist.**
(DOC)

**S1 File.**
(DOCX)

## Acknowledgments

We thank Lori Rosman (Lead informationist, Welch Medical Library, Johns Hopkins University) for her advice on the literature search strategy.

## Author Contributions

**Conceptualization:** Vignesh Chidambaram, Nyan Lynn Tun, Waqas Z. Haque, Petros C. Karakousis, Panagis Galiatsatos.

**Data curation:** Vignesh Chidambaram, Nyan Lynn Tun, Waqas Z. Haque, Ranjith Kumar Sivakumar, Amudha Kumar, Angela Ting-Wei Hsu, Izza A. Ishak, Aqsha A. Nur, Samuel K. Ayeh, Emmanuella L. Salia, Ahsan Zil-E-Ali, Muhammad A. Saeed, Ayu P. B. Sarena, Bhavna Seth, Muzzammil Ahmadzada, Eman F. Haque, Pranita Neupane, Kuang-Heng Wang, Tzu-Miao Pu, Syed M. H. Ali, Muhammad A. Arshad, Lin Wang, Sheriza Baksh.

**Formal analysis:** Vignesh Chidambaram, Marie Gilbert Majella, Sheriza Baksh.

**Investigation:** Vignesh Chidambaram, Angela Ting-Wei Hsu.

**Methodology:** Vignesh Chidambaram, Marie Gilbert Majella, Lin Wang, Sheriza Baksh, Panagis Galiatsatos.

**Supervision:** Sheriza Baksh, Petros C. Karakousis, Panagis Galiatsatos.

**Visualization:** Ranjith Kumar Sivakumar, Amudha Kumar.

**Writing – original draft:** Vignesh Chidambaram, Nyan Lynn Tun, Marie Gilbert Majella, Ranjith Kumar Sivakumar, Amudha Kumar, Izza A. Ishak, Emmanuella L. Salia, Bhavna Seth, Petros C. Karakousis, Panagis Galiatsatos.

**Writing – review & editing:** Vignesh Chidambaram, Marie Gilbert Majella, Amudha Kumar, Izza A. Ishak, Sheriza Baksh, Petros C. Karakousis, Panagis Galiatsatos.

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
