## [Decision Letter · Decision Letter 0]

7 Oct 2020

PONE-D-20-24192

Factors Associated with Disease Severity and Mortality among Patients with Coronavirus Disease 2019: A Systematic Review and Meta-Analysis

PLOS ONE

Dear Dr. Galiatsatos,

Thank you for submitting your manuscript to PLOS ONE. After careful consideration, we feel that it has merit but does not fully meet PLOS ONE’s publication criteria as it currently stands. Therefore, we invite you to submit a revised version of the manuscript that addresses the points raised during the review process.

We look forward to receiving your revised manuscript.

Kind regards,

Girish Chandra Bhatt, MD, FASN

Academic Editor

PLOS ONE

Journal Requirements:

4. Please include your tables as part of your main manuscript and remove the individual files.

Please note that supplementary tables should be uploaded as separate "supporting information" files

5. Please include captions for your Supporting Information files at the end of your manuscript, and update any in-text citations to match accordingly. Please see our Supporting Information guidelines for more information: http://journals.plos.org/plosone/s/supporting-information

Reviewers' comments:

Reviewer's Responses to Questions

**Comments to the Author**

1. Is the manuscript technically sound, and do the data support the conclusions?

Reviewer #1: Yes

2. Has the statistical analysis been performed appropriately and rigorously? 

Reviewer #1: Yes

3. Have the authors made all data underlying the findings in their manuscript fully available?

Reviewer #1: Yes

4. Is the manuscript presented in an intelligible fashion and written in standard English?

Reviewer #1: Yes

5. Review Comments to the Author

Reviewer #1: Comments

A very interesting systematic review article to read and it highlights some important findings. Some concerns that need to addressed.

1. “We did not include articles uploaded in the preprint servers, as they are not peer reviewed and the findings may not be reliable”

The pre-prints serve as a source of unpublished data and if the authors have planned to not include unpublished data, they should state it

2. “Data extraction was performed independently by at least two of the authors (VC, NT, WH, MM, RS, AK, AH, II, AN, SA, ES, MS, AS, KW, and TP) and conflicts were resolved by a consensus between two authors (MM and VC)” Were these two authors acting as arbitors ? If so please mention it.

3. “The risk assessment for bias for all the studies included in this review was performed using the Newcastle-Ottawa quality assessment scale (NOS) for observational studies” Please mention that you used the NOS for cohort studies

4. The protocol registration number is missing. If it was registered apriori?

5. As regarding the heterogeneity, the authors have been trying to adjust for heterogeneity but haven’t really looked in for the reasons of the same. Highlighting the reasons of heterogeneity will be important for the readers and users of this systematic review.

6. PLOS authors have the option to publish the peer review history of their article (what does this mean?). If published, this will include your full peer review and any attached files.

Reviewer #1: No

---

## [Author Response · Author response to Decision Letter 0]

11 Oct 2020

To the Editor:

1) We have added the corresponding author’s ORCID ID.

2) Tables were added to the manuscript.

3) We have added captions for the figures, and are embedded in the manuscript.

4) All supplemental material in the attached PDF have tables and figures with titles.

To Reviewer #1:

A very interesting systematic review article to read and it highlights some important findings. Some concerns that need to addressed.

1. “We did not include articles uploaded in the preprint servers, as they are not peer reviewed and the findings may not be reliable”

The pre-prints serve as a source of unpublished data and if the authors have planned to not include unpublished data, they should state it

Response: We have re-worded the statement as such:

We only included article published in peer-reviewed academic journals; we did not include articles uploaded in the preprint servers, as they are not peer reviewed and the findings may not be reliable(10).

2. “Data extraction was performed independently by at least two of the authors (VC, NT, WH, MM, RS, AK, AH, II, AN, SA, ES, MS, AS, KW, and TP) and conflicts were resolved by a consensus between two authors (MM and VC)” 

Were these two authors acting as arbitors ? If so please mention it.

Response: An important question to address; these authors were acting as arbitors and mentioned as such in the manuscript.

3. “The risk assessment for bias for all the studies included in this review was performed using the Newcastle-Ottawa quality assessment scale (NOS) for observational studies” 

Please mention that you used the NOS for cohort studies

Response: We have added that language into the manuscript.

4. The protocol registration number is missing. If it was registered apriori?

Response: The protocol was not registered.

---

## [Editor Report · Decision Letter 1]

19 Oct 2020

Factors Associated with Disease Severity and Mortality among Patients with Coronavirus Disease 2019: A Systematic Review and Meta-Analysis

PONE-D-20-24192R1

Dear Dr. Galiatsatos,

We’re pleased to inform you that your manuscript has been judged scientifically suitable for publication and will be formally accepted for publication once it meets all outstanding technical requirements.

Kind regards,

Girish Chandra Bhatt, MD, FASN

Academic Editor

PLOS ONE
---

## [Editor Report · Acceptance letter]

10 Nov 2020

PONE-D-20-24192R1 

Factors Associated with Disease Severity and Mortality among Patients with COVID-19: A Systematic Review and Meta-Analysis 

Dear Dr. Galiatsatos:

I'm pleased to inform you that your manuscript has been deemed suitable for publication in PLOS ONE. Congratulations! Your manuscript is now with our production department. 

Kind regards, 

on behalf of

Dr. Girish Chandra Bhatt 

Academic Editor

PLOS ONE